# An Evaluation of the Impact of Databases on End-of-Life Embodied Carbon Estimation

**Augustine Blay-Armah** [1], **Ali Bahadori-Jahromi** [1,*] , **Anastasia Mylona** [2], **Mark Barthorpe** [3] **and Marco Ferri** [3]

1 Department of Civil Engineering and Built Environment, School of Computing and Engineering, University of West London, London W5 5RF, UK; 21490520@student.uwl.ac.uk
2 Research Department, The Chartered Institution of Building Services Engineers (CIBSE), London SW12 9BS, UK; amylona@cibse.org
3 LIDL Great Britain Ltd., 19 Worple Road, London SW19 4JS, UK; mark.barthorpe@lidl.co.uk (M.B.); marco.ferri@lidl.co.uk (M.F.)
* Correspondence: ali.bahadori-jahromi@uwl.ac.uk

**Abstract:** The growing awareness of the need to minimise greenhouse gas (GHG) and mitigate climate change has resulted in a greater focus on the embodied carbon (*EC*) of construction material. One way to ensure the environmental impact of building activities is minimised to a reasonable level is the calculation of their *EC*. Whilst there are a few studies investigating the role of embodied carbon factor (ECF) databases on the accuracy of *EC* calculation from cradle to gate, very little is known about the impact of different databases on the end-of-life (EoL) *EC* calculation. Using ECFs derived from the UK Department for Business, Energy and Industrial Strategy (BEIS), the Royal Institute of Chartered Surveyors (RICS) default values and the Institution of Structural Engineers (IStructE) suggested percentages for different elements of a building's lifecycle stages, this study presents the impact of different data sources on the calculation of EoL *EC*. The study revealed that a lack of EoL ECFs databases could result in a significant difference of about 61% and 141% in the calculation of *EC*.

**Keywords:** embodied carbon; embodied carbon factor databases; end of life; life cycle assessment; recycling

## 1. Introduction

Globally, there is a growing awareness of the need to limit greenhouse gas (GHG) and mitigate climate change. The most recent annual UN climate change conference—Conference of the Parties (COP26) summit held in the United Kingdom and attended by delegates from regional, national, and international levels—underscores the global commitment to reduce GHG.

Construction is amongst the leading sectors contributing to global economic growth whilst having a huge adverse impact on resource consumption, GHG emission, solid waste generation, and global warming [1–3]. Globally, buildings are responsible for 39% of carbon emissions, up to 36% of energy and natural resources consumption and almost 50% of the solid waste disposed of in landfills [2,4]. Previously, however, most effort was focused on reducing the operational carbon of buildings rather than the embodied carbon (*EC*) [5,6]. Without a comparable effort on the reduction of the *EC* of buildings and construction materials, efforts towards net-zero-emission buildings and GHG emissions reduction to mitigate climate change would be compromised. According to UK Green Building Council (UKGBC) [7], *EC*, also referred to as carbon capital, can be defined as follows: 'the total greenhouse gas emissions generated to produce a built asset'. This encompasses carbon emission emitted during extraction of raw material, processing and manufacturing of building material, transporting and assembling of building product, and deconstruction or demolition of building material and its disposal. Furthermore, Akbarnezhad and Xiao [8] observed that the whole-life *EC* can be reduced by considering the carbon footprint implications of the chosen strategy to deal with the end of life (EoL)

of a building. Thus, the whole lifecycle of a building should be taken into account in an attempt to achieve net-zero buildings and reduce the built environment's impact on climate change. Whilst past research carried out by Mohebbi et al. [9] on the role of *EC* coefficients databases in the estimation of *EC* accuracy at the cradle to gate revealed that the use of a comprehensive database compared to a generic database can lead to a 35.2% reduction of carbon emissions, very little is currently known about the impact of different databases on EoL phase *EC* calculation. Reliable and credible databases provide useful information for a transparent calculation of carbon emissions.

As a building approaches its EoL phase, the material stocks and associated *EC* will be released, and therefore, it is crucial to select appropriate strategies to manage different materials and components of the building as well as the accompanied *EC* invested in them [8]. The adoption of the concept of recycling had long been identified as one of the management strategies to limit the harmful environmental impact at the EoL stage while achieving material efficiency by acting as a feedstock to close the material loops [10,11]. Therefore, the purpose of this paper is to explore the effect of different databases on the estimation of EoL *EC*. It seeks to examine the currently available databases in the UK and how they impact EoL *EC* calculations.

## 2. Literature Review

### 2.1. Recycling

One of the main strategies to deal effectively with demolished building materials or components at the end of a building's useful service life is recycling [8,12]. Recycling is defined as the process of converting construction and demolition waste into new material [13]. While the process of recycling may result in carbon emissions, it is encouraged as an alternative strategy to raw material extraction to deal with construction and demolition waste [14]. It is therefore essential these emissions are assessed when selecting recycling as a strategy for carbon emission minimisation. Factors influencing the quantity of carbon emission during the recycling process are the materials being recycled and the level of technological advancement of the recycling process [15]. Akbarnezhad and Xiao [8] claimed that the initial *EC* invested in building materials during the production and construction processes can be released during the process of recycling at the end of the building's useful life. The authors, therefore, suggested that in the absence of data, the initial carbon can be used to estimate the emissions of recycling structural elements or materials.

Notwithstanding, the environmental benefits of recycling demolished building materials at end of the service life in an attempt to reduce carbon emissions have been documented [5,16]. For instance, Hopkinson et al. [16] observed that due to the challenges to reclaiming concrete, much emphasis has been on recycling instead of reuse. Yuan [17] further highlighted that recycling diverts waste materials from being sent to landfills and removes the need for virgin materials. Reducing waste production through recycling is a key factor in material resource efficiency. Wu et al. [5] asserted that recycling has now become the common EoL management strategy for concrete, and the proportion of concrete being recycled keeps increasing year on year. For instance, according to the British Ready-Mixed Concrete Association [18], in the UK, approximately 90% of concrete can be recycled or recovered. Hence, recycling as an EoL management strategy promotes effective resource utilisation by extending the life span of the building materials, thus improving the building and construction sector in an ecologically friendly way. Nevertheless, there is no current study that examines the impact of different databases on EoL-phase embodied carbon calculation.

### 2.2. Embodied Carbon and Operational Carbon

The total carbon emissions generated during the whole life cycle of a building are usually classified into operational emissions and embodied emissions. Operational carbon emissions are the result of energy used during the use phase of the building and represent approximately 28% of the global energy consumption [2], while embodied carbon is the

total amount of emissions of GHG emitted over the life cycle of the building accounting for almost 11% of the energy used [6]. The initial *EC* is a product-based emission that occurs prior to the construction of the building and involves raw materials extraction, manufacturing, and transporting of products to a construction site. The construction *EC* is related to the construction stage of the building, whilst the recurring *EC* emission is associated with maintenance, replacement, deconstruction, demolishing, and disposal of the building materials. In this paper, however, *EC* is limited to reoccurring only. This is because this study only concentrates on the *EC* emissions during the deconstruction or demolition of building materials and their disposal.

In the past, a large amount of effort has been concentrated on optimising operational carbon. However, due to the whole lifecycle of the buildings, carbon emissions extend beyond the use of the building. Additionally, with the race to net-zero carbon intensified, there could be no operational carbon for buildings in the future, and all carbon emissions will be assigned to embodied carbon [19]. Therefore, the key to reducing the impact of buildings on climate change is to minimise *EC* emissions, and the whole lifecycle of a building should be taken into consideration in order to reap long-term benefits.

### 2.3. Life Cycle Assessment

One way to ensure the environmental impact of construction activities is minimised to a reasonable level, whilst providing the needed economic and social infrastructure, is the application of the life cycle assessment (LCA) tool [20,21]. LCA is a dynamic tool that can be employed to assess the consumption of raw material and energy, carbon emission, and waste associated with the whole useful life of a product or a system [21,22]. It is a well-recognised quantification tool that allows the comparison of different materials used in a project and their choice of management strategies [10].

As a multipurpose and useful environmental impact assessment tool, LCA can be employed at a single-product level to calculate the carbon emission arising from the product across its useful lifespan or to determine the environmental impacts of several products and processes over their entire life cycle [23,24]. It is an internationally accepted approach that provides a standardised methodological basis for quantifying carbon emission, consumption of energy, depletion of natural resources, and other environmental impacts throughout the whole lifecycle of buildings [25–27].

The LCA methodology follows the four-stage framework (goals and scope definition, life cycle inventory (LCI), LCA, and interpretation recommended by [22]. The EN 15978 [28] for buildings environmental performance assessment provides guidelines for the sustainability of construction activities. Figure 1 illustrates the structure and definition of stages in the life cycle of buildings.

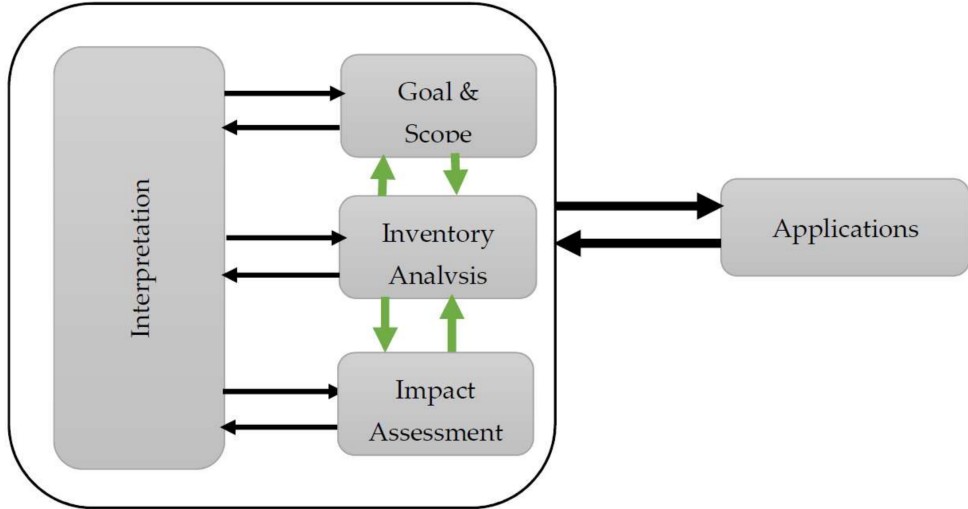

**Figure 1.** Life cycle assessment stages, reproduced [22].

The scope of this study is limited to the calculation of the *EC* of buildings. LCA scope is EoL (C1–C4), as shown in Figure 2. Module C1 encompasses all the activities and processes in the deconstruction or demolition of the building at the end of its useful service life and includes carbon emissions associated with the use of equipment, fuel consumption, and related emissions. Module C2 activities include the transportation of the deconstructed or demolished building materials to the storage site for reuse, recycling plants, waste treatment plants, or landfill sites. The carbon emissions associated with C2 depend on the mode of transport and fuel consumption as well as the distance travelled. Module C3 includes all the activities associated with the waste treatment plant, while Module C4 encompasses the carbon emissions of the processes associated with the final disposal of building materials.

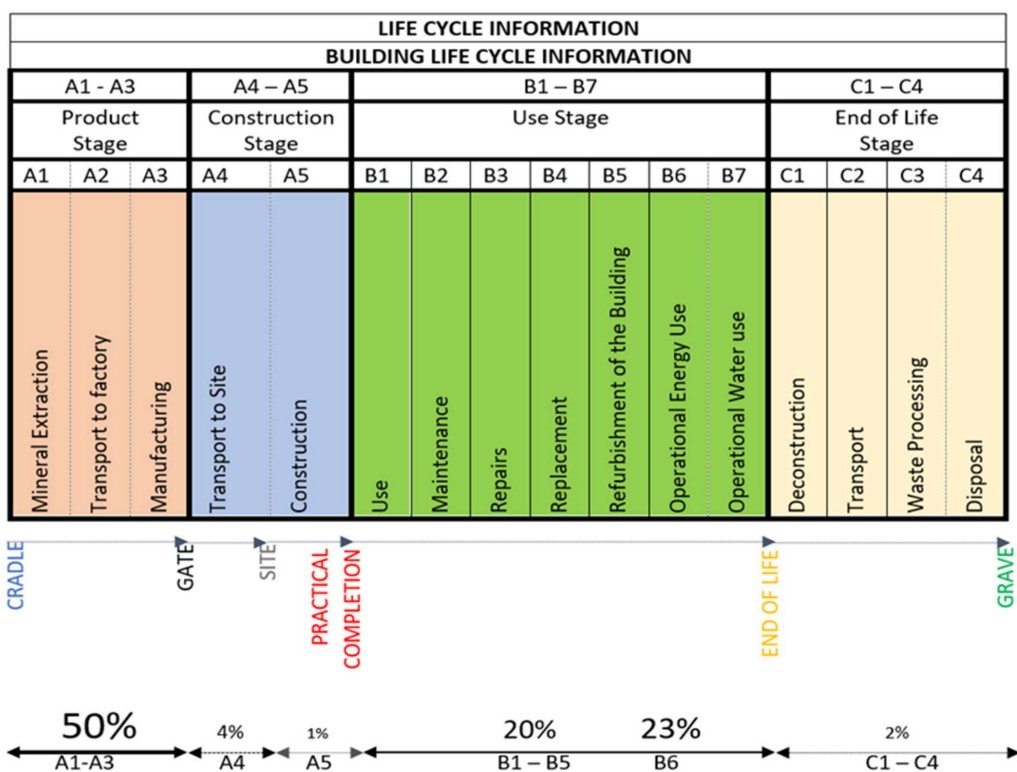

**Figure 2.** Life cycle stages and modules with split carbon emission across all building elements, adapted [29].

The life cycle of a building is divided into stages, and the carbon emission boundary of each stage is determined. The carbon source of each stage is quantified according to the material quantities and the emission factors of various materials that are determined. Regarding the carbon footprint of building material, however, the whole life cycle or an isolated life cycle stage can be selected as a boundary system. The ISO 14067 [30] specifies the assessment method for the carbon footprint of a product by providing some specific requirements on the selection of system boundary and the simulation of other phases but cradle to gate. This document specifies that the construction phase can be used as the system boundary for carbon emission calculation only when:

- Information on a specific stage such as EoL of the products is unavailable, and reasonable scenarios cannot be modelled, or
- The other phases have insignificant impact on the calculation of the carbon emissions of the product.

Therefore, for certain materials/products' whole lifecycle estimation, the EoL stage may be left out only if its results are considered insignificant. Hence, the need to investigate the impact of ECF databases on the calculation of EoL embodied carbon.

### 2.4. Databases and Embodied Carbon Factors State of the Art

One of the crucial data requirements in the assessment of *EC* of buildings is the materials and components emission coefficients or embodied carbon factors (ECFs). Accurate carbon emission coefficients are vital for reliable *EC* estimation. These factors can be obtained from various secondary sources including national data, industry data, commercial lifecycle database, PAS 2050 compliant carbon footprint, aggregated or derived from the literature, and Environmental Product Declarations (EPDs). The accuracy and reliability, however, differ from one database to another. Consequently, EN 15978 [29] requires that the most recently updated data be used and verified with the provisions of EN 15804 [31,32]. According to Gervasio and Dimova [33], emission coefficients can be obtained from two main sources—generic and specific. Generic refers to datasets that are based on material quantities production and construction processes specific to the geographical area where the structure is built. These data sources may include national data and data derived from the literature. Therefore, any amendments in these details can have a significant impact on the results of the estimation. Generic databases should be used with caution, as they cannot be assumed to possess similar features as those of other geographic regions where both material production and construction processes differ [34]. The data should fit conditions of the geographical area of production and construction procedures. Hence, the use of a specific data source can enhance the accuracy of lifecycle assessment.

Specific data, on the other hand, are supplied by producers and manufacturers in the form of EPDs and externally validated to certify the environmental impact of the product in accordance with the requirements of BS EN 15804 [35]. Additionally, the EPDs production process must meet the standard of ISO 14044 [22]. The goal of EPDs is to share building materials or products' environmental impacts with users [32]. According to Gelowitz and McArthur [36], EPDs provide freely available environmental data. Ibáñez-Forés et al. [37] have also observed that one of the important features of EPDs is to act as a valuable source of transparency to understand the environmental impacts of construction materials and processes. In addition, the availability of EPDs affords assessors more certainty in their findings; therefore, they are noted as an effective way of transmitting products' environmental performance [38]. The aforementioned demonstrate that EPDs are a useful data source to gain an accurate picture of building materials' environmental performance which can aid the assessment of *EC*.

However, EPDs are not mandatory for all the lifecycle stages except the A1–A3 boundary [35]. Although EPDs are currently a progressively growing source of the environmental database for the built environment, they are still limited in number [38]. The available literature suggests there are various reasons why EPDs are not currently well-positioned to aid whole lifecycle assessment and for comparison [38,39]. Andersen et al. [38] suggested that EPDs are presently accessible for a limited number of building materials, whilst Hunsager et al. [39] pointed out the challenge of the distrust of users concerning inadequate transparency and validity. Additionally, Bhat and Mukherjee [40] discussed the issue of reliability in EPDS. The authors argued that inconsistency of results because of uncertainty can greatly affect data quality. Therefore, these drawbacks limit not only the quality and use but also the comparability of EPDs.

Currently, in the UK, the Inventory of Carbon and Energy (ICE) is recognised as the most reliable database for carbon factors. It was developed in the late 1990s by the University of Bath and summarises embodied carbon coefficients for most common construction materials [41]. This database contains more than 500 building materials commonly used in construction but provides only cradle-to-gate carbon factors. Therefore, the number of available databases providing ECFs for the EoL phase is limited. In this study, materials for which no EPDs could be obtained were inventoried from available databases and literature with similar production conditions of the UK.

## 3. Methodology

### 3.1. Calculation of End-of-Life Embodied Carbon

The study adopts a process based LCA methodology (where the physical flow of all aspects of building materials can be identified and traced) to establish the *EC* calculations during the EoL phase. Two sets of input data are required to enable the calculation of embodied carbon. These are material quantities and ECFs.

### 3.1.1. Embodied Carbon Factors

As noted earlier, one of the vital data requirements in the assessment of *EC* of buildings is the ECFs. However, given the limited nature of databases providing ECFs for the estimation of EoL *EC* in the UK, ECFs were obtained from the following data sources.

The first data source is a national database created by the formerly Department for Energy and Climate Change [42], now Department for Business, Energy and Industrial Strategy (BEIS). This database contains over 40 ECF classifications including construction. Construction material classifications for both cradle to gate and EoL management (waste disposal) included aggregates, asbestos, asphalt, bricks, concrete, construction average, insulation, metals, soils, mineral oil, plasterboard, tyres and wood.

The Royal Institute of Chartered Surveyors (RICS) suggests that if more specific data are unavailable, derived or aggregated information can be used [43]. Based on monitored case studies of demolition activities in the UK, the RICS manual provides average values to assist the calculation of the carbon emissions arising during the EoL of buildings. In accord with ISO 14044 [22] requirements, the RICS manual recommends that the selected data should be the latest and representative of the geographic location of the project as well as technologically up-to-date. Thus, these aggregated values represent the second data source.

Similar to the RICS manual, the IStructE guide recommends that in the absence of more specific data, default percentages for various lifecycle phases can be applied to aid the calculation of *EC* (see Figure 2) [29]. The guide suggests 2% for the management of the EoL phase of a building, 50% for the product stage, 5% for the construction phase, and 43% for the use phase. The third data source in this study is the IStructE guide.

### 3.1.2. Material Quantities

Another important parameter in the calculation of *EC* is the material quantities. For simplicity of measurement, the material quantities may be expressed in mass, volume or area [44]. Hence, the calculation of an *EC* of each material for the life cycle was determined by Equation (1):

$$EC_i = \sum_i (Qmaterial_{,i} \times ECF_i) \tag{1}$$

where $EC_i$ refers to the total embodied carbon of material$_{,i}$, Q is the total quantity of material$_{,i}$, and ECF represents the embodied carbon factor of material$_{,i}$.

The system boundary is the macro phase of EoL which can be further divided into C1-to-C4 subsections; however, C3 and C4 are mutually exclusive in this study. Accordingly, the total amount of *EC* of the building at this phase was estimated using Equation (2):

$$EC_{eol} = EC_1 + EC_2 + EC_3 \tag{2}$$

The embodied carbon emission associated with demolition was estimated by:

$$EC_1 = \sum_m (Qmachinery_{,m} \times ECFmachinery_{,m}) + \sum_e (Qenergy_{,e} \times ECFenergy_{,e}) \tag{3}$$

where $EC_1$ refers to embodied carbon, m is the type of machinery for on-site operation, and e is the type of energy used.

The embodied carbon emission associated with transporting dismantled building elements or components to the storage site for reuse or demolished materials off site to the recycling plant was estimated by:

$$EC_2 = \sum i(Qtrans_{,i} \times ECFtrans_{,i}) \tag{4}$$

where $EC_2$ represents the carbon emissions associated with transporting the building element$_{,i}$ or the demolished materials$_{,i}$.

The embodied carbon emission associated with processing demolished waste was calculated by:

$$EC_3 = \sum_i(Qwp_{,i} \times ECFwp_{,i}) \tag{5}$$

where $EC_3$ represents the carbon emissions associated with the processing of waste materials$_{,i}$.

To facilitate the comparison, the total *EC* of a data source was divided by the initial total *EC* based on the calculation of cradle to grave for the initial construction of the entire building to give a change in difference (DA) for implementing the EoL strategy (recycling) for the entire building materials, as determined by Equation (6):

$$DAi = \frac{Vi}{ECi} \times 100\% \tag{6}$$

where *V* represents the total embodied carbon from a data source for the entire building material$_i$.

From Equation (6), a higher percentage change between data sources indicates a significant difference in EoL embodied carbon estimation between the datasets.

The system boundary of this study was set to the entire building, including substructure and superstructure, but excluding the external works and heating and ventilation system. An LCA conducted focuses on carbon emission associated with the EoL phase of a typical supermarket building in the UK. The building comprises a steel frame and a composite panel external wall and was simulated using the BIM application. It is a single-story structure, with an area of 2500 m$^2$ as illustrated in Figure 3.

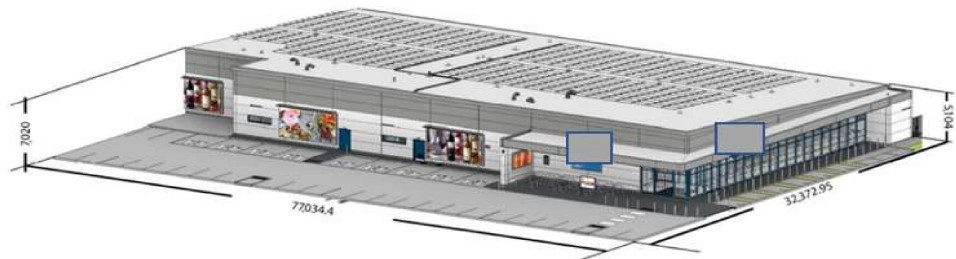

**Figure 3.** A 3D simulation of a typical supermarket building.

The extracted building materials were categorised in accordance with the classification system suggested by the RICS [43], as shown in Table 1.

The structural components obtained included ceilings and ceiling finishes, curtain wall, doors, floor surfaces and finishes, roof, structural frame, foundation, external wall, internal wall and finishes and windows

### 3.1.3. Background and Scenario

In order to make the comparison objectively and intuitively along with ensuring reliability and accuracy in the results, ECFs used in this study were sourced from the UK databases to ensure geographical and regional conditions of production procedures, construction practices, energy consumption, and building design characteristics were met. To reiterate, the sources of ECFs used in this study were as follow: (i) BEIS, (ii) RICS and (iii) IStructE.

**Table 1.** Building Elements and Structural Components of a 2500 m$^2$ Supermarket Building.

| Building Element | Structural Element and Component |
|---|---|
| Substructure | Foundation including foundation wall and floor slab |
| Superstructure | Structural frame: roof beams, columns and tie beams<br>Roof: steel profile system on tapered insulation<br>Upper floor: concrete<br>Stairs and ramps |
| External envelope | External walls: steel frame and insulated cladding panels, concrete and glazed curtain walling.<br>Windows and metal external doors |
| Interiors | Internal walls: metal framed plasterboard, concrete, blocks and paster and timber.<br>Internal finishes: floorings—ceramic tiles, vinyl and paint.<br>Ceilings—tiles, concrete, plasterboard and timber.<br>Metal doors |

The quantities of building material were obtained from the BIM model. Using Equation (1) and ECFs from the ICE database, the lifecycle embodied carbon was calculated as displayed in Table 2.

**Table 2.** Material Weight and Calculated Initial Carbon Based on a 2500 m$^2$ Supermarket Building.

| Material | Weight (tonne) | Initial Embodied Carbon (tonneCO$_2$e) |
|---|---|---|
| Aluminium | 11.7 | 111.8 |
| Bricks | 30.1 | 14.8 |
| Concrete | 1055.6 | 285.0 |
| Glass | 0.7 | 5.2 |
| Insulation | 307.6 | 1147.5 |
| Plastic | 1.3 | 8.9 |
| Plasterboard | 40.4 | 4.8 |
| Steel | 283.2 | 2701.3 |
| Tiles | 60.4 | 8.9 |
| Timber | 0.1 | 27.1 |
| **Total** | **1791.1** | **4315.3** |

Three hypothetical scenarios were constructed to enable effective comparison. First, it was assumed the entire building would be demolished at end of its 30-year useful life. Second, to ensure effective comparison, a 100% recycling of all demolished materials was assumed, although the UK has 90% and 96% recovery rates for concrete and steel, respectively [18,43]. A 100% recycling refers to a scenario wherein all the demolished materials are recycled [10]. Third, a road transport distance of 50 km fully laden was assumed.

Within the BEIS database, Bricks, Concrete, Insulation and Steel have the same ECF of 0.989. Glass, Plastic and Plasterboard have an ECF of 21.294, while Aluminium and Tiles were assigned a construction average of 0.989.

For RICS, an average value of 3.418 was assigned to all types of materials. This figure was obtained by summing up a default figure of 3.400 for the carbon emissions occurring during either on-site or off-site deconstruction and demolition activities (C1), 0.005 for carbon emissions related to the transportation of demolished materials (C2) and 0.013 for carbon emissions associated with waste processing (C3).

Finally, the IStructE guide recommends 2% of the total initial carbon, as indicated in Figure 2.

## 4. Results and Discussion

The study examined the effect of databases on EoL management strategy in which all the generated demolished building materials were directed to recycling. As noted earlier,

the simulated building was steel frame and precast concrete, and therefore, concrete and steel made up a major proportion (74.7%) of the building material, with concrete being 58.9%, and steel being 15.8%. Table 3 shows an overview of the calculated *EC* for the different materials in the 2500 m$^2$ supermarket building using the three data sources. The single most striking observation emerging from the data comparison was the totals.

i.   Comparison between BEIS and RICS: the results from these two data sources showed that the share of BEIS' total of the embodied carbon was less than half of RICS.

ii.  Comparison between BEIS and IStructE: comparison between these two data sources totals revealed that the share of IStructE was about 30 time less than that of BEIS data source.

iii. Comparison between RICS and IStructE: in totals, compared with the IStructE guide, the RICS source gave the higher contribution of carbon emissions, up to 70.9 times more than IStructE.

**Table 3.** Calculated Embodied Carbon for Each Database Based on a 2500 m$^2$ Supermarket Building.

| Material | Material: Weight (tonneCO$_2$e) | Initial *EC* (tonneCO$_2$e) | BEIS (tonneCO$_2$e) | RICS (tonneCO$_2$e) | IStructE (tonneCO$_2$e) |
|---|---|---|---|---|---|
| Aluminium | 11.7 | 111.8 | 11.6 | 40.1 | 2.2 |
| Bricks | 30.1 | 14.8 | 29.8 | 102.9 | 0.3 |
| Concrete | 1055.6 | 285.0 | 1044.0 | 3608.0 | 5.7 |
| Glass | 0.7 | 5.2 | 14.4 | 2.3 | 0.1 |
| Insulation | 307.6 | 1147.5 | 304.3 | 1051.5 | 23.0 |
| Plastic | 1.3 | 8.9 | 28.3 | 4.6 | 0.2 |
| Plasterboard | 40.4 | 4.8 | 857.1 | 137.6 | 0.1 |
| Steel | 283.2 | 2701.3 | 280.1 | 968.0 | 54.0 |
| Tiles | 60.4 | 8.9 | 59.8 | 206.6 | 0.2 |
| Timber | 0.1 | 27.1 | 1.8 | 0.3 | 0.5 |
| **Total** | **1791.1** | **4315.3** | **2631.2** | **6121.9** | **86.3** |

The differences among the three datasets for concrete are highlighted in Table 3. As can be seen from the results, the IStructE guide provided the least carbon emission, followed by the BEIS data source and the RICS. A comparison between the BEIS data source and the RICS manual revealed a significant difference of 2564.0 tonneCO$_2$e between the two sources, representing more than twice the BEIS result. The most surprising aspect of the data is the difference of 3602.3 tonneCO$_2$e between RICS and IStructE sources, whilst a difference 1038.3 tonneCO$_2$e was revealed between the BEIS data source and the IStructE guide.

From the data in Figure 4, it is apparent that there were significant differences among the three datasets for steel. The most striking result emerging from the data is the RICS data source result of almost 1000 tonneCO$_2$e. The difference between the RICS manual (highest) and the IStructE guide (lowest) was about 914 tonneCO$_2$e.

Turning now to the percentage difference, data from Table 3 and Figure 4 reveal stark variations among the three sources of data for all building materials. For aluminium and steel, the BEIS ECF accounted for 10% of the initial whole-lifecycle *EC*, while the RICS default value represented 36%. In addition, 202% was recorded for using ECF from the BEIS data source for bricks, with 698% for the RICS default value. For concrete, the RICS data source showed 1266% share of the initial whole-lifecycle *EC*, and the BEIS data source revealed a share of 366% (see Figure 4). A share of 277% was found for glass with the ECFs from the BEIS data source, whilst RICS showed a share of 44%. For insulation, the use of the RICS default value showed a share of 92% in the whole-lifecycle embodied carbon, while the ECF from BEIS database provided a 27% share. There was an increase in *EC* for recycling plastic, the BEIS data source indicating 319%, and RICS 51%. An increase of 17,745% in *EC* was recorded when using ECF from the BEIS source for recycling plasterboard, while a rise of 2848% was obtained when using the RICS default value. There was a rise of 668% in *EC* for recycling tile using ECF from BEIS, and an increased of 2309% when using the RICS

manual. Finally, the use of ECF from the BEIS database for timber showed 7% increased share in *EC*, while RICS default value indicated a 1% rise.

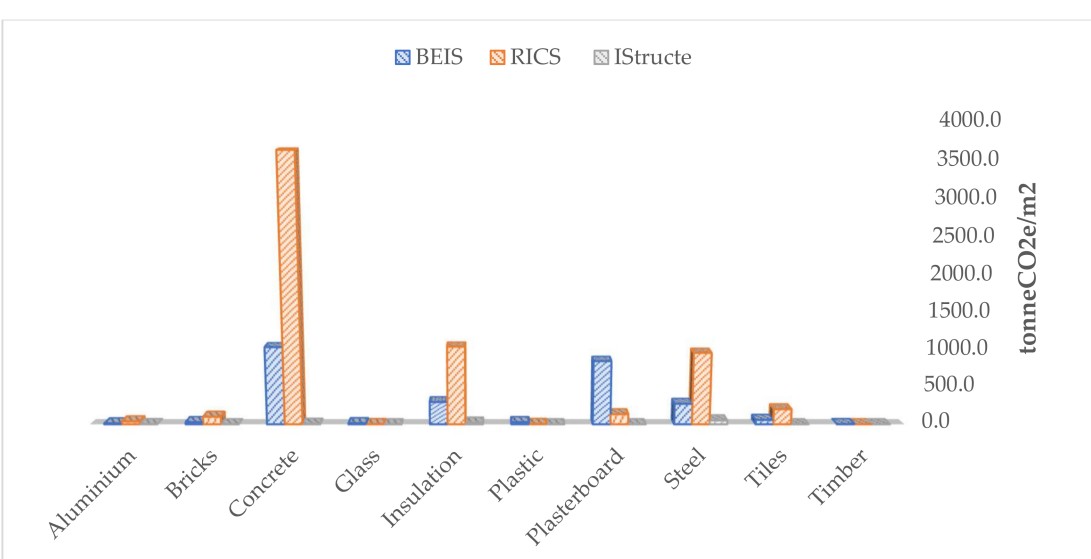

**Figure 4.** Comparison of Data Sources Embodied Carbon of Building Materials Based on a 2500 m² Supermarket Building.

Together, these results provide important insights into the impact of databases on the estimation of *EC* at the EoL phase of buildings. Whilst the IStructE data source was based on 2% of the total initial whole-lifecycle embodied carbon, the BEIS data source and the RICS manual showed a percentage change of almost 61% and 141%, respectively, with an 80% difference between the two data sources.

A comparison among individual materials within the project was carried out to demonstrate how sourcing carbon factors from different databases can impact the results of *EC* calculation. Given the variance in the results among the three databases, which might be due to the limited availability of carbon factors for EoL analysis, none of the databases appeared to provide a desirable result. Since the choice of the data source for carbon factors can greatly influence the reliability of the results, LCA assessors should proceed with caution.

These results further suggest that more needs to be done to improve the shortage of a detailed database for ECFs at the EoL phase of a building. While there are more detailed databases available to source ECFs for other phases of the life cycle, particularly cradle to gate, there are limited data sources for the EoL. In the absence of detailed databases, default ECFs values have to be used in the calculation of *EC* at the EoL stage. For instance, in their cradle-to-gate study to assess the impact of databases on the accuracy of embodied carbon estimation, Mohebbi et al. [9] found that the choice of database for ECFs had an insignificant impact on the results for concrete material. The authors, however, attributed their findings to the availability of more detailed databases and subsequent ECFs at this phase of the building's life cycle. Contrarily, this study has shown that limited databases can significantly impact the *EC* calculation outcome [45]. This lack of data can be attributed to the comparatively long building useful lifespan and uncertain nature of the EoL processes or activities [46]. Although the use of default national EoL values and assumptions enable EoL modelling and reduce the time required in performing LCA, the lack of data sources providing ECFs hampers the reliability of assessment results.

In their study to model the end-of-life phase of buildings and allocate benefits and burdens, Mirzaie et al. [46] found that information in generic databases concerning EoL activities such as deconstruction, transportation, recycling, and disposal, in many cases, is

little or non-existent. Considering that concrete forms the larger proportion of this supermarket building, recycling concrete can further reduce the life cycle carbon emissions [30].

An attempt to reduce the *EC* of a building starts from the design stage, and conducting LCA is key to selecting materials with low carbon intensity. The estimation of *EC*, therefore, provides a basis for material selection not only at the design phase of a new project but also during construction and beyond. However, the shortage of a detailed database for ECFs at the end of the useful life of building materials requires extra time and labour on the part of the assessor. This can serve as a disincentive in carrying out LCA. On the other hand, a detailed data source can offer alternatives, thereby reducing the chances of resorting to derived or aggregated ECFs during estimation to ensure reliability and confidence in the results.

Besides, the primary goal of this study was to explore the impact of sources of databases for ECFs on EoL-phase *EC* calculation to mitigate climate change. There have been a few studies carried out in this area. Studies involving EoL are mostly based on assumptions. This study, however, was based on a real-case project, with real data used for the calculation of *EC*. The findings of this research can contribute to consistent collection of ECFs to create databases, which could guarantee reliability in the estimation of *EC* in the effort to mitigate the global climate change. While the study is also intended to encourage change in the built environment, the availability of better data can empower the construction sector to reduce carbon emission on a significant scale. Furthermore, the assessment methodology can be adopted in other studies around the globe to guide the environmental impact assessment and demonstrate the amount of *EC* in similar structures.

## 5. Conclusions

The aim of *EC* estimation is to provide useful, reliable, transparent and credible information for designers, developers, investors and other stakeholders to make informed decisions to mitigate climate change. The selection of a database for ECFs directly influences the results upon which decisions are made. Understanding the impact of databases on *EC* estimation can greatly contribute to the whole lifecycle of embodied carbon reduction.

The study demonstrated that the lack of EoL ECFs databases compared to more comprehensive databases for cradle to gate could result in a significant difference between about 61% and 141%, thereby overestimating the result for implementing the necessary EoL strategy. This is crucial, as it can misinform designers, investors and other stakeholders in their attempts to reduce carbon emissions and to mitigate climate change. The key to ensuring reliability and credibility in *EC* estimation results is the accuracy of ECF databases. However, there is a lack of a reliable database for ECFs for buildings' EoL phase, impacting negatively in the calculation of whole-life *EC*. The lack of EoL datasets can lead to time waste and increase inconsistency between life cycle assessors. It is therefore recommended to address this gap in life cycle inventory databases in the future to ensure accurate calculation of carbon emissions at the embodied phase of buildings. The availability of data sources for ECFs for EoL can reduce inaccuracies in the calculation of *EC* but also encourage more LCA to be conducted.

Nonetheless, it is noteworthy to point out the limitations of this study, which can affect its practicality. The assumption of a 100% recycling rate for all building materials considered in this study does not correspond to a real situation. In the UK, for instance, the main materials in this supermarket building—concrete and steel—have recovery rates of 90% and 96%, respectively. Notwithstanding, the findings of this study should facilitate not only decision making regarding the availability and accuracy of ECF data sources, particularly EoL of building materials, but also credible and reliable whole-lifecycle carbon emissions calculation at the embodied phase of buildings.

**Author Contributions:** Conceptualisation, A.B.-J., A.M., M.B. and M.F.; Research, 3D modelling, calculation and formal analysis, A.B.-A.; Writing—original draft preparation and editing, A.B.-A.; Writing—review and editing, and supervision, A.B.-J., A.M., M.B. and M.F. All authors have read and agreed to the published version of the manuscript.

**Funding:** This research received no external funding.

**Data Availability Statement:** All data generated from this study are available within the text of this manuscript.

**Conflicts of Interest:** The authors declare no conflict of interest.

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
