# Peer review of "An Evaluation of the Impact of Databases on End-of-Life Embodied Carbon Estimation"

_sustainability, doi:10.3390/su14042307_

Round 1

Reviewer 1 Report

In the present research, the authors try to evaluate the impact of databases on End-of-Life embodied carbon estimation. The paper exhibits some results, but there are some questions.

  1. The authors titled the paper as “An Evaluation of the Impact of Databases on End-of-Life Embodied Carbon Estimation”. Based on the title, the database should be one the most important factor. However, the authors have not clearly defined the databases separately and demonstrated their key data. That influences the following understanding.
  2. In the experimental, the authors set the end of life for building for four stages. And they also give the equation of ECeol=EC1+EC2+EC3+EC4. However, they just provide the detailed calculation of EC1, EC2 and EC3. Then, it wonders what the EC4 is? If it could be neglected, it need not list in the equation above. If it is possible, it should be detailed.
  3. In the content, the authors use the kgCO2e and tonneCO2e to describe the similar things. That is some confusing. The authors are suggested to unify them.
  4. In the content, the authors provide the calculated embodied carbon for each material according to the different standards. However, it is still wondered what the details are?
  5. In the experimental, the authors provide the material weight and calculated initial carbon. It wonders whether these data are based on the building of 2500 m2 supermarket? If that is, the authors should describe them on the table caption.
  6. In the Table4, the authors provide the calculated data. However, what is the different between the Table 4 and Table 3, except the unit?
  7. The authors should pay more attention on the spelling to avoid the errors.

Reviewer 2 Report

The article is interesting and addresses a current and important research topic. The study presents the impact of different data sources on the calculation 20 of EoL embodied carbon. However, the article needs improvement.

1.The novelty of the presented approach in comparison to other studies is not clearly demonstrated. This should be better explained.

2.The article should be carefully edited. For example, the paragraph, line 166-172, is puzzling: The Materials and Methods should be described with sufficient details to allow others to replicate and build on the published results. Please note that the publication of your manuscript implicates that you must make all materials, data, computer code, and protocols associated with the publication available to readers. Please disclose at the submission stage any restrictions on the availability of materials or information. New methods and protocols should be described in detail while well-established methods can be briefly described and appropriately cited.

Round 2

Reviewer 1 Report

The authors have revsied the manuscirpt according to the suggestions. Now it is improved and could be accepted.